# Recruitment and Retention Strategies Used in Dietary Randomized Controlled Interventions with Cancer Survivors: A Systematic Review

**DOI:** 10.3390/cancers15174366

**Published:** 2023-09-01

**Authors:** Samantha J. Werts, Sarah A. Lavelle, Tracy E. Crane, Cynthia A. Thomson

**Affiliations:** 1Mel and Enid Zuckerman College of Public Health, University of Arizona, Tucson, AZ 85724, USA; cthomson@arizona.edu; 2University of Arizona Cancer Center, University of Arizona, Tucson, AZ 85724, USA; 3College of Agriculture and Life Sciences, University of Arizona, Tucson, AZ 85721, USA; sarahlavelle@arizona.edu; 4Miller School of Medicine, University of Miami, Miami, FL 33136, USA; tecrane@med.miami.edu; 5Sylvester Comprehensive Cancer Center, University of Miami, Miami, FL 33136, USA

**Keywords:** enrollment, attrition, validity, reliability, diet, cancer survivors

## Abstract

**Simple Summary:**

The purpose of this systematic review is to evaluate the quality of recruitment and retention methodologies in the context of diet-related intervention trials among cancer survivors. Findings suggest investigators are meeting reporting guidance for recruitment; however, reporting of retention methods and rates is less consistent, raising concern as to the interpretation of study findings. There is a need for researchers to consistently report retention methods and rates to inform best practices and enhance the rigor of future diet intervention trials in cancer survivors.

**Abstract:**

Background: The purpose of this review was to systematically evaluate the quality of reporting of recruitment and retention methods in diet-related intervention trials among cancer survivors. Methods: A systematic search of five databases in Spring 2023 identified dietary intervention randomized controlled trials with a minimum of 50 cancer survivors, an intervention of at least eight weeks, and at least six months of study duration. Outcomes investigated include methodologic description and reporting of recruitment and retention rates. Results: Seventeen trials met inclusion criteria. Recruitment methods included cancer registry and clinician referral, hospital records, flyers, and media campaigns, and were reported in 88.2% of studies. Eleven of 17 studies (64.7%) met a priori recruitment goals. Eleven studies identified an a priori retention goal and seven met the goal. Retention goals were met more often for studies of less than one year (71.4%) versus greater than one year (50%), and for studies with remote or hybrid delivery (66.7%) versus only in-person delivery (50%). Conclusions: Recruitment goals and methods are frequently reported; reporting of retention methods and goals is limited. Efforts are needed to improve reporting of retention methods and rates to inform best practices and enhance the rigor of future dietary intervention trials.

## 1. Introduction

It has been estimated that as of 2022, over 18 million people in the United States were cancer survivors, and this number is expected to surpass 22 million by 2030 [1]. Cancer survivors are at an increased risk of second primary cancers along with other diet-related chronic illnesses such as diabetes, osteoporosis, and cardiovascular disease [2]. Incorporating healthy eating behaviors plays a key role in reducing comorbidities, improving overall health, promoting a longer lifespan, and supporting higher quality of life for cancer survivors [2]. As the population of those who have lived with cancer expands, leading organizations have proposed guidelines for dietary behaviors to promote survival, reduce co-morbidity, and increase quality of life. The World Cancer Research Fund (WCRF) and the American Institute for Cancer Research (AICR) recommend maintaining a healthy weight during adulthood as well as consuming a diet with a high daily intake of whole grains, vegetables, fruit, and legumes, and restricting consumption of sugar-sweetened beverages, processed foods, red meat, and alcohol [3,4]. The American Cancer Society (ACS) and American Society of Clinical Oncology (ASCO) provide similar recommendations for cancer survivors with additional advice to limit saturated fat intake [5,6].

Since diet is a factor in morbidity and mortality outcomes among cancer survivors, numerous interventions have been conducted to enhance adherence to dietary guidance for survivorship [7,8,9,10,11]. Several published reviews acknowledge the beneficial effects of these dietary interventions on health outcomes for cancer survivors [7,8,9]. To ensure the scientific rigor of these trials, adequate and appropriate methodologies and transparency of reporting must be followed [12]. Two of the important aspects that impact the validity of study results are the recruitment and retention of participants in clinical research. In fact, concerns related to the appropriate interpretation of findings from randomized controlled trials arise when recruitment (accrual) falls below statistically powered estimates, or when drop-out rates (attrition) exceed a priori estimates [13]. Differential recruitment or drop-out rates by study randomization assignment is an additional concern that reduces trial integrity and rigor [14,15]. Reporting of a priori sample size is a necessary step for other researchers to determine if the study reached recruitment and retention goals and thus amply tested the proposed hypothesis.

Recruiting adequate participants to randomized controlled trials and retaining them for the duration of the intervention requires coordinated and strategic efforts. Some methods of recruitment include searching cancer registries for potential participants and sending invitation letters to those eligible, promoting the study through in-person and online networks, and advertising through media outlets, including traditional television, radio, and print [16]. Retaining participants in clinical trials can be difficult, often related to burden on the participant. Burden may include frequency of measurements, travel time and cost to attend study visits, time required to adhere to the intervention, time to prepare food and measure for self-reported data, time to perform self-assessment (tracking food intake or steps), family time commitments, and the social burden with family and friends not participating in the intervention [8,17,18,19]. It is crucial that researchers report the reasons why participants exit the study, thus providing key insight to inform the balancing of participant burden with research rigor in future research trials. The application of CONSORT (CONsolidated Standards of Reporting Trials) diagrams to randomized controlled trial outcome publications is one step journals have taken to increase transparency in reporting participant engagement throughout a trial [20].

The purpose of this systematic review is to evaluate the quality of reporting in the context of diet-related intervention trials among cancer survivors and to describe recruitment and retention methods previously reported. Our focus is on published randomized controlled dietary intervention trials among four cancer survivor populations: breast, prostate, colorectal, and lung and bronchus. This review focuses on accrual and attrition outcomes of these trials, and descriptions of recruitment and retention methods, offering insight into approaches that may enhance reporting and promote optimal recruitment and retention in dietary trials among cancer survivors.

### Aims

The aims of this systematic review are to (1) describe recruitment and accrual rates and methods used to recruit cancer survivors into randomized, controlled dietary intervention trials, and to (2) describe retention and attrition rates and methods used to promote retention of cancer survivors enrolled in randomized controlled dietary intervention trials.

## 2. Methods

The conduct and reporting of this systematic review adhere to the Preferred Reporting Items for Systematic Reviews and Meta-Analyses (PRISMA) guidelines [21]. The protocol was registered with PROSPERO, the International Prospective Register of Systematic Reviews (CRD42018070396).

### 2.1. Search Strategy

The following five electronic databases were searched for relevant articles in January 2023: PubMed, CINAHL, Cochrane Central Register of Controlled Trials, Embase, and PsychINFO. Search strategies were developed with assistance from a librarian and adapted for each database using standardized terminology. The search string was developed for PubMed and then translated to the other databases. Keywords or Medical Subject Heading (MeSH) terms included in the electronic database search comprised (1) diet, (2) cancer survivors—specifically breast, prostate, colorectal, lung and bronchus, and (3) randomized controlled trials and controlled clinical trials (see Appendix A for example of the search strategy). Filters included human adults and peer-reviewed articles published between 2013 and 2023 in English.

### 2.2. Eligibility Criteria

This systematic review evaluates dietary behavior change and diet-inclusive weight loss intervention randomized controlled trials, including pilot and feasibility studies with a minimum of 50 participants at the time of randomization, at least 8 weeks of active intervention, and at least 6 months of total expected participation. Eligible studies included adult (18+ years old) breast, prostate, colorectal, and lung and bronchus cancer survivors. Studies with other cancer types were allowed if one of these primary cancer types was included in the study. Outcomes investigated in this review include recruitment and retention methods and rates, which required the presence of a CONSORT flow diagram of participant engagement numbers, attrition rates, and reasons participants discontinued participation in a trial [20]. Trials still in active recruitment were ineligible. 

### 2.3. Study Selection and Data Extraction

A single individual (SW) performed the database search, and all citations were exported to EndNote X9 for data management. Duplicate citations were removed following the process described by Bramer et al., 2016 [22]. Two reviewers (SL, SW) then independently dual-screened the titles and abstracts of all publications to assess eligibility criteria. Articles that did not meet the inclusion criteria were excluded, and any articles for which there was a discrepancy between reviewers moved on to full-text screening for further review. Full-text articles were screened independently by two reviewers (SL, SW) to assess eligibility. Non-relevant articles were excluded, and conflicting votes were resolved by a third reviewer (CT).

Data were extracted independently by SL and SW using a data extraction form developed by SL, SW, TC, and CT. SL and SW screened the full-text articles and reference lists for protocol or design articles for the selected studies. Data collected from each study or protocol article included first author, study title, publication year, country where the study was conducted, study name, study design, cancer type and stage, a priori sample size, intervention type, mode of delivery, length of intervention and follow-up, methods of recruitment and retention, time necessary for recruitment, number recruited, number screened, number eligible, number randomized, and number retained. Retention rate was calculated as the percent of the number of participants who completed study follow-up divided by the number randomized. Attrition rate was calculated as the percent of the number of participant withdrawals divided by the total number randomized.

## 3. Results

Articles were identified through database searches (n = 2000) in January 2023. References were imported into EndNote X9, and deduplication procedures were followed resulting in a total of 1396 articles for title and abstract screening. Full-text articles were screened (n = 83), and after exclusion of articles based on a priori criteria, 20 articles were included for data analysis [10,23,24,25,26,27,28,29,30,31,32,33,34,35,36,37,38,39,40] (Figure 1). Three of the articles described the same study, resulting in a total of 17 study cohorts (Table 1).

### 3.1. Study and Participant Characteristics

Each study cohort represented a randomized controlled trial with 14 identified as a 2-arm randomized controlled trial [10,24,25,27,28,30,32,33,34,35,37,38,40,42], one a 3-arm trial [36], one a 4-arm trial [29], and one a 2X2 factorial design [31]. Two studies included a wait-list control [24,40], and two were described as pilot feasibility trials [29,42]. Studies were conducted throughout the world, with seven conducted in the United States [24,27,29,36,37,38,40], two conducted in Australia [10,42], and one each conducted in North America (United States and Canada) [30], England [28], Germany [32], the Netherlands [33], Scotland [34], Northern Ireland [35], Italy [25], and Hong Kong [31]. Average age of enrolled participants across studies ranged from 49.9 to 71.0 years, with 29.4% (n = 5) of studies enrolling a participant population with an average age of 65 years or greater. Of the ten studies which reported race and ethnicity, five (50%) enrolled a sample which was greater than 90% White [10,29,30,36,42], and three (30%) enrolled a sample greater than 70% White [23,39,40].

Seven of the 13 studies which reported education level for their participants reported samples with greater than 70% of participants having at least some college education [23,29,31,36,37,39,40]. Most studies (n = 16) focused on cancer survivors of a specific cancer type, while one included all cancer types [33]. Most studies focused on breast cancer (n = 10, 58.8%) [10,24,25,27,30,32,36,37,38,42], four (23.5%) studies enrolled prostate cancer survivors [28,29,34,35], and two (11.8%) recruited colorectal cancer survivors [31,40]. Sample sizes across the included studies varied greatly. The average total randomized study sample size was n = 289.2 (SD: 368.3; range: 50–1542), with ten studies enrolling a sample size over n = 100 [10,25,29,30,31,32,33,36,37,39], and seven studies enrolling a sample size of n = 100 or less [23,27,28,34,35,40,42].

The review aimed to include diet and weight loss interventions; four studies focused solely on dietary change [25,27,37,40], eleven combined diet and physical activity [10,24,28,29,30,31,34,35,36,38,42], while the remaining two included diet and other behavior changes including smoking cessation, sleep, or psychosocial well-being [32,33]. Study intervention duration ranged from 12 weeks to 5 years with most (64.7%, n = 11) interventions lasting between 12 weeks and 6 months [27,28,29,32,33,34,35,36,37,40,42]. Most studies followed patients for six months (47.1%, n = 8) [27,28,29,32,33,35,40,42] and were conducted remotely (52.9%, n = 9) [10,29,30,32,33,35,36,40,42] with telephone or text-based coaching calls, online modules, or mailed study materials. Six studies included a hybrid model [24,28,31,34,37,38] incorporating some in-person sessions along with home-based materials, and two were conducted entirely in-person [25,27].

### 3.2. Recruitment Results

Of the 15 studies that reported their target accrual, eleven (73.3%) met or exceeded their recruitment goals [10,26,28,29,32,33,35,37,39,40,42]. The four (26.7%) studies that did not meet their recruitment goals [23,30,31,36] identified lack of adequate accrual time, challenges with ineligibility criteria, and the loss of funding or the end of the funding cycle as reasons for not meeting their accrual goals. Two studies did not report their target accrual goals [25,34]. Overall, studies reported a lack of interest as the primary reason potentially eligible participants did not enroll in the study, followed by lack of access to transportation for in-person sessions or assessments, and lack of time.

The average target sample size for the eleven studies which met their a priori accrual goals was 212.4 ± 192.9 participants and 674.7 ± 985.2 participants for the four studies which did not meet accrual goals, though this difference was not significant (*p* = 0.42). Accrual time across studies ranged from three months to four years, with the average accrual time being 19.9 ± 11.8 months. There was no difference in average accrual time between studies which met their target accrual goals and those which did not.

All nine studies utilizing a remote delivery model reported their recruitment goals, and seven (77.8%) met these goals [10,29,32,33,35,40,42]. One of the in-person studies [26] and three of the five hybrid-delivered studies [28,37,39] reported and met their recruitment goals. Of the eleven studies that incorporated both diet and physical activity behavior change goals, ten reported their accrual goal [10,24,28,29,30,31,35,36,38,42], and six (60.0%) met their goal [10,28,29,35,39,42]. Of the four studies which only targeted dietary change, three reported their goal, and all three (100%) met these goals [26,37,40]. The two studies which incorporated multiple behavior changes reported and met their recruitment goals [32,33]. Ten of the eleven studies with an intervention duration of six months or less reported their accrual goals [27,28,29,32,33,35,36,37,40,42], and nine (90.0%) met these goals [27,28,29,32,33,35,37,40,42]. Five of the six studies with an intervention duration longer than six months reported their accrual goals [10,23,30,31,38], and only two (40.0%) met these goals [10,39]. Six of the nine (66.7%) breast cancer studies that reported their accrual goals met their goal [10,26,32,37,39,42], compared to 100% of the prostate cancer studies (n = 3) [28,29,35], and 50% of the colorectal cancer studies (n = 1) [40].

Most studies utilized a clinic recruitment model (70.6%, n = 12) with either provider referral to the study or review of the electronic health record to identify eligible patients [10,27,28,30,31,32,33,34,35,36,38,40]. Eight studies (47.1%) recruited participants from a cancer or tumor registry [10,24,25,29,34,36,38,42], four (23.5%) recruited from cancer support groups [25,32,38,40], and three (17.6%) recruited using media-based advertisements (i.e., social media, television, radio, etc.) [25,32,38]. One study reported utilizing clinicaltrials.gov to support study recruitment [36], and one reported recruiting at community-based events [38]. One study did not report recruitment methods or sources [37]. Clinical recruitment methods were the most commonly utilized recruitment methods among studies which met their recruitment goals. The four studies which did not meet their recruitment goals only utilized either a clinical recruitment method or recruitment from a cancer registry, while the eleven studies that did meet their goals also recruited participants from community-based events, media, and cancer support groups.

### 3.3. Retention Results

Eleven (64.7%) of the 17 studies identified a priori retention goals [10,24,27,28,29,31,33,35,38,40,42]. Seven studies met these goals [10,23,26,29,33,35,40]. Over a third of the studies did not report retention goals [25,30,32,34,36,37], and two did not report any participant retention methods [25,29]. Retention goals were met more often in studies less than one year in duration (71.4%) compared to those longer than one year (50.0%) and in studies with remote or hybrid delivery (71.4%) compared to in-person delivery (50.0%). Of the eight diet- and physical activity-focused studies which reported their target retention [10,24,28,29,31,35,38,42], four (50.0%) met these goals [10,23,29,35]. Of the two diet-focused studies which reported their retention goals, both met these goals [27,40]. One of the two multi-behavior change studies identified and met its retention goal [33]. Three of the five (60.0%) studies enrolling breast cancer survivors identified and met their retention goals [10,23,27]. Two of the three prostate cancer-focused trials met their retention goals [29,35], and one of the two colorectal cancer-focused studies met its goal [40].

Eleven trials (64.7%) reported a five percent or greater difference in retention by study arm [10,25,30,31,32,33,34,36,37,40,42]. These eleven studies were more likely to have not reported a retention goal (45.5%) than the six studies which saw a less than five percent retention difference by study arm (16.7%) [23,25,27,28,35,39]. Studies which did not report a retention goal were more likely to see greater attrition in the intervention arm compared to the control arm, both at the end of the intervention (83.3%) and the end of follow-up (83.3%). For the eleven studies that reported their retention goals [10,24,27,28,29,31,33,35,38,40,42], there was no difference in retention goal acquisition between studies that saw greater control participant attrition versus intervention participant attrition.

Commonly reported reasons for participant attrition included loss to follow up or noncompliance, lack of willingness to be wait-listed for the intervention or join a control arm, primary outcome event (i.e., death, recurrence, new disease), and inability to attend in-person sessions or assessments. Retention methods included regular study contact through text, phone call, or email (47.1%) [10,28,30,33,34,35,37,38,40]; participant compensation or travel allowance (23.5%) [10,24,31,36]; personalized coaching or lifestyle recommendations (23.5%) [27,32,37,42]; and flexible scheduling (11.8%) [10,31]. Consistent contact was the most commonly employed method by studies achieving retention goals. Compared to studies which did not meet their identified retention goals, studies that did meet their retention goals more commonly included delayed delivery of intervention materials to the control group, rapport building and personalized messaging, or tailored materials.

## 4. Discussion

This review highlights recruitment and retention of cancer survivors in dietary randomized controlled trials published over the past decade. In general, reporting of recruitment methods and study target accrual was high. Although all studies reported overall participant attrition rates, a key finding of this review is the overall limited reporting of protocol-specified participant retention goals (30% of trials provided) and related retention methodology used to achieve retention goals. Of note, trials of shorter duration and with remote or hybrid intervention delivery more commonly reported achieving recruitment goals and maintaining high retention. This has implications in terms of sustaining engagement in lifestyle behavior programs, a factor that will drive long-term health promotion, and related oncology care outcomes in cancer survivors [43].

In terms of recruitment strategies, oncology clinic-based recruitment was the most common approach. This has implications in terms of the timing of interventions as cancer survivors are encouraged to move back to primary care within five years of treatment [44]; thus, recruitment in oncology care may bias early survivors and promote recruitment for individuals with higher access to care. This may indirectly contribute to inequities in access to these trials and related lifestyle health coaching programs and is a problem that has been noted in previous work. In fact, the lack of diversity in trials has been a major impetus for the community outreach and engagement role in comprehensive cancer center care [45], wherein the National Cancer Institute called to action the need for broader catchment area enrollment in trials that represent the diversity of people with cancer to reduce cancer health disparities [46]. Current data suggest diversity is not achieved in most oncology research [47]. Similar to previous work, here we show that every trial recruiting across race and ethnic groups resulted in over 82%, on average, being non-Hispanic White (NHW) race and ethnicity, despite the higher cancer mortality rates and likelihood for diagnosis with advanced disease in Blacks and Hispanics compared to NHW [1]. Further, Black and Hispanic cancer survivors, as compared to NHWs, report lower adherence to dietary guidance for cancer survivors [48,49], supporting a greater need for programming focused on attaining a healthful diet.

Our results highlight the need for more transparency in reporting retention goals and methodologies in lifestyle behavior oncology research. We identified that remote- or hybrid-delivered interventions are most likely to meet retention goals. Similar to what has been concluded in previous work, this suggests continued efforts to develop effective programming using web-based and telephone resources [50]. Additionally, studies which reported protocol-specified retention goals and strategies were less likely to see differences in attrition by study arm compared to those that did not report. Importantly, 80% of the studies which did not report a priori retention goals saw greater attrition in the behavior change condition compared to usual care or control. This is compared to the studies which did report retention goals, of which less than 10% saw greater attrition in the intervention arm. Differential drop-out rates between study conditions have implications for trial rigor and validity, potentially biasing study results and therefore dietary recommendations for survivors [15]. Systematic and protocol-driven retention methodology should be developed early in the planning stages of a randomized, controlled trial and reported clearly to support interpretation of trial results.

Several resources and recommendations for improving recruitment and retention in clinical trials are available. The NIH-funded Recruitment Innovation Center was established in 2016 to help increase enrollment of diverse participants in clinical trials and offers support to investigating teams to ensure timely completion of trials and with statistical power to make valid recommendations for care [51]. To overcome barriers for trial participation, the community-engaged recruitment model [52] and the patient-centered recruitment and retention model [53] have been developed. Suggested approaches include cultural tailoring of intervention materials, consideration and adaptation for participants with geographic barriers, offering digital or remote alternatives to program implementation, and providing travel compensation and flexibility [54,55], which our results supported as effective in achieving enrollment and retention goals. Building community partnerships is a well-described and relevant strategy to improve accessibility to trials [54,56], including diet trials in survivors, which was not well utilized in the included studies, as only one study reported recruiting at community-based events [39].

This review has several strengths, including being the first to systematically address recruitment and retention in completed dietary-controlled trials for cancer survivors. The systematic approach to trial selection and data collection limits the risk of bias. Study eligibility criteria for this review captured trials with moderate or high levels of research rigor (i.e., randomized trials, greater than 50 participants at randomization, at least six months of follow-up, etc.). Our results inform effective strategies to improve recruitment and retention that are applicable for advancing trial participation in future diet behavior change programming and in turn supporting completion of more rigorous and evidence-based clinical trials.

With a limited number of trials included and from a wide range of geographic locations, comparisons between recruitment and retention methodology must consider heterogeneity in trial contexts and settings. A potential limitation to these data is publication bias. Studies with statistically significant findings are more likely to be published than those with non-significant or negative findings or those which were ended early due to futility, a potential result of low enrollment or high attrition in a study [57]. Another potential threat to the validity of study results and a limitation for our study is healthy subject bias. Participants experiencing greater symptom or comorbidity burden related to their cancer treatment may be more likely to drop-out or be lost to follow-up, biasing trial results towards the healthiest participants or those with greater access to resources for cancer-related care [58]. Future research should also explore other factors that may impact survivor engagement and adherence to an intervention such as individual affect, cultural and social influences, dietary preference, and age. Clear reporting of patient-reported outcomes is necessary to understand if attrition and resulting missing data are related to poorer outcomes, methodological issues, or lack of interest in the intervention [59].

## 5. Conclusions

Achieving and retaining study participants in dietary intervention trials among cancer survivors are important to the interpretation of study findings. The most common recruitment methods utilized for studies which achieved their a priori accrual goals included clinical recruitment and provider referral or recruitment from a cancer registry. Studies with a duration of six months or less, and studies with remote or hybrid delivery models were most likely to have met their accrual goals. Our findings suggest investigators are meeting reporting guidance for transparency in study recruitment; however, reporting of retention methods and rates is less common and inconsistent, raising concerns as to the interpretation of study findings. Studies which met their identified retention goals most frequently reported regular contact and interaction between study staff and participants. This indicates that rapport building is an important factor for survivor engagement and continuation with a trial. Efforts to improve retention reporting should be enhanced, especially for trials of longer duration.

## Figures and Tables

**Figure 1 cancers-15-04366-f001:**
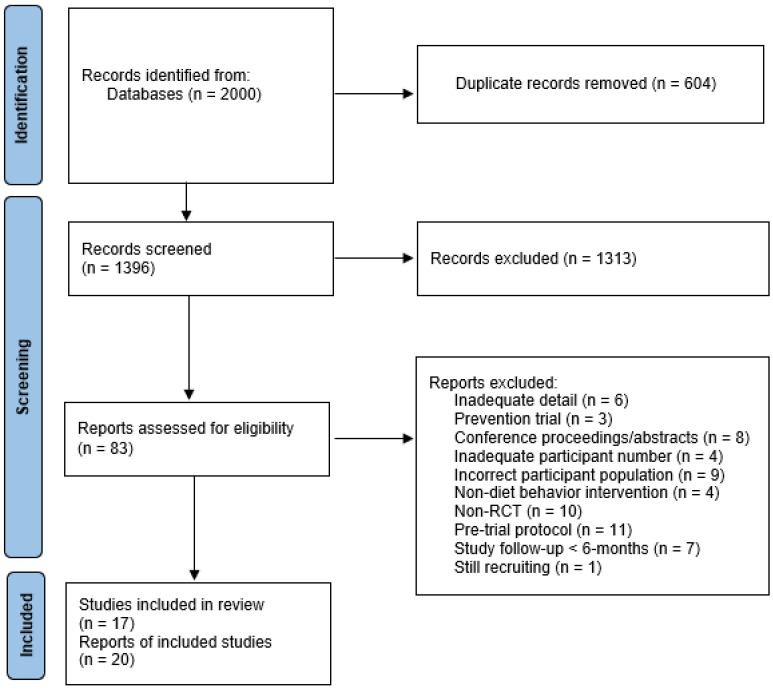
Systematic review flow-diagram. Modified from [41].

**Table 1 cancers-15-04366-t001:** Study characteristics, recruitment, and retention data for randomized, controlled dietary trials conducted with cancer survivors over the past ten years (n = 17).

Study Name, First Author, Year, Country	Study Design, a Priori Sample Size Inclusion Criteria	Mean Age (SD), Enrollment by Sex, Race, Education	Intervention Behavioral Targets, Duration	Mode of Intervention Delivery	Recruitment Source and Time Frame for Accrual	Recruitment Methods, Number Contacted, and Number Randomized	Did Study Meet Recruitment (Accrual) Goal?	Retention Methods (Rates by Time-Point and Group, if Available)	Did Study Meet Retention (Attrition) Goal?
Harvest for Health, Birmingham Breast Cancer Survivors (BBCS), Bail, 2018; and Cases, 2016 (United States) [23,24]	Two-arm wait-list control, crossover RCT Target accrual: n = 100 (1) BC (2) post-primary treatment (3) eating <5 servings of vegetables and fruits/day (4) exercising <150 min/week (5) ≥1 physical function limitation	60.2 (11.1) years 100% female 73.2% (n = 60) White 83.0% (n = 69) ≥ some college	Diet and PA 24 months	Home-based and in-person, Master gardener led gardening intervention, gardening journals/notebooks	University of Alabama cancer registry, self-referrals August 2013–May 2014 (10 months)	Cancer registry, n = 1279 mailed study invitations, n = 1128 survivors contacted, n = 194 screened, n = 82 deemed eligible 32.6% response rate n = 82 randomized (n = 44 intervention, n = 38 delayed intervention)	No, 82% of accrual target Reason: lack of time to recruit equal number of participants from surrounding 5 counties, many rural county participants were already gardening and ineligible	Methods: Participant compensation, USD 500 worth of gardening tools Total: 97.6% (n = 80) Intervention: 95.4% (n = 42) Control: 94.7% (n = 36)	Yes, goal was 80% retention
Diet and Androgen-5 (DIANA-5) Trial, Bruno, 2021 (Italy) [25]	Two-arm RCT A priori target accrual not defined (1) invasive BC (2) post-primary treatment within 5 years (3) 35–70 years	52.0 (8.5) years 100% female Not reported Not reported	Diet 5 years	In person classes (4) and 10 food-centric meetings	Tumor registries, oncology units, breast cancer screening units, breast cancer support groups, media January 2008–December 2010 (3 years)	No details n = 1542 randomized (n = 769 intervention, n = 773 control)	Not reported	Methods: Not reported 1 year: Total: 87.1% (n = 1344) Intervention: 89.6% (n = 689) Control: 84.7% (n = 655)	No retention goal identified
*¡Cocinar para su salud!*, Bernard-Davilla, 2015; and Greenlee, 2015 (United States) [26,27]	Two-arm RCT Target accrual: n = 70 (1) BC, stage 0–III (2) >3-months post-treatment (3) Hispanic descent	56.6 (9.7) years 100% female 40.0% (n = 28) White 38.6% (n = 27) ≥ some college	Culturally based diet intervention 12 weeks, 9 sessions Extended follow-up at 6 months	In-person, community based: culturally based group cooking classes, grocery shopping field trip, nutrition education	Columbia University Medical Center (CUMC) Breast Oncology Clinic January 2011–March 2012 (1 year and 3 months)	Clinical recruitment, n = 405 potentially eligible (n = 142 refused to participate, n = 37 unable to contact), n = 111 screened, n = 102 eligible, n = 70 enrolled n = 70 randomized (n = 34 intervention, n = 36 control)	Yes, 100% of accrual target	Methods: Culturally adapted and tailored materials provided in Spanish 3 months: Total: 96% (n = 67) Intervention: 91.2% (n = 31) Control: 100% (n = 36) 6 months: Total: 87% (n = 61) Intervention: 88.2% (n = 30) Control: 86.1% (n = 31)	Yes, expected 15% attrition at 6 months
Effect of lifestyle intervention in men with advanced PC on ADT, Bourke, 2014 (England) [28]	Two-arm single blind RCT Target accrual: n = 100 (1) PC on ADT for locally advanced and metastatic PC (2) taking ADT for ≥6 months	71 years, range: 53–87 years 100% male Not reported Not reported	Diet and PA 12 weeks Extended follow-up at 6 months	In-person: supervised exercise program and small-group healthy eating seminars, home-based: nutrition advice pack	Outpatient clinics 2008–2012 (4 years)	No details on number screened, n = 135 deemed eligible n = 100 randomized (n = 50/arm)	Yes, 100% of accrual target	Methods: Extended supervision and contact during follow-up 12 weeks: Total: 85% (n = 85) Intervention: 86% (n = 43) Control: 84% (n = 42) 6 months: Total: 68% (n = 68) Intervention: 70% (n = 35) Control: 66% (n = 33)	No, only expected 25% attrition at 6-months
TrueNTH Community of Wellness, Chan, 2020 (United States) [29]	Four-arm pilot RCT Target accrual: n = 200 (1) PC (2) physician clearance	Median 70 years old, IQR: 65–75 years 100% male 92.6% (n = 187) White 92.6% (n = 187) ≥ some college	Diet and PA 3 months Extended follow-up at 6 months	Level 1: information on website, Level 2: information on website and exercise videos, Level 3: information on website, exercise videos, Fitbit, text messaging, Level 4: information on website, exercise videos, Fitbit, text messaging, two 30-min telephone calls with a dietician and/or exercise trainer	Hospital cancer registry databases, the Cancer of the Prostate Strategic Urologic Research Endeavor registry, clinics 2017–2019 (13 months)	Potentially eligible participants identified from the cancer registry mailed study letter, n = 6406 mailed study information, n = 292 expressed interest, n = 240 screened, n = 220 deemed eligible n = 202 randomized Level 1 n = 49 Level 2 n = 51 Level 3 n = 50 Level 4 n = 52	Yes, 101% of total accrual target (did not meet goal of n = 50 per arm)	Methods: not reported 3 months: Total: 82.7% (n = 167) Level 1: 79.6% (n = 39) Level 2: 84.3% (n = 43) Level 3: 86% (n = 43) Level 4: 80.8% (n = 42) 6 months: Total: 77.2% (n = 156) Level 1: 77.6% (n = 38) Level 2: 78.4% (n = 40) Level 3: 76% (n = 38) Level 4: 76.9% (n = 40)	Yes, expected 20% attrition at 3 months and 36% attrition at 6-months
The LISA Trial, Goodwin, 2014 (North America) [30]	Multicenter, 2-arm RCT Target accrual: n = 2150 (1) postmenopausal women (2) diagnosed with T1-3N0-3M0 BC in the previous 36 months (3) currently using letrozole^®^	60.4 years (7.8) for mail-based vs. 61.6 years (6.7) for intervention 100% female 95.6% (n = 323) White Not reported	Diet and PA 24 months	Home-based, mailed information on healthy living, workbook, and individualized telephone-based lifestyle coaching intervention in English or French	16 Canadian and American participating medical centers May 2007–January 2010 (2 years 8 months)	Clinical recruitment, n = 682 identified as potentially eligible, n = 546 consented and screened, n = 338 deemed eligible n = 338 randomized (n = 167 mail-based, n = 171 intervention)	No, 15.7% of accrual target Reason: enrollment was terminated due to loss of funding	Methods: Mailed and telephone reminders 6 months: Total: 93.4% (n = 316) Mail-based: 92.8% (n = 155) Intervention: 94.2% (n = 161) 12 months: Total: 85.5% (n = 289) Mail-based: 88% (n = 147) Intervention: 83% (n = 142) 18 months: Total: 82.5% (n = 279) Mail-based: 86.2% (n = 144) Intervention: 78.9% (n = 135) 24 months: Total: 78.1% (n = 264) Mail-based: 78.4% (n = 131) Intervention: 77.8% (n = 133)	No retention goal identified
Moving Bright, Eating Smart, Ho, 2013 (Hong Kong) [31]	Multicenter, 2 × 2 factorial RCT Target accrual: n = 224 (n = 56 per cell) (1) CRC (2) within one-year post-primary treatment	65.2 years (10.1), range: 25 to 86 years 36.8% female (n = 82), 63.2% male (n = 141) Not reported 87.4% (n = 195) ≥ some college	Diet and PA 12 months Extended follow-up at 24 months	Motivational interviews with registered dietitian, telephone calls, mailed newsletters and pamphlets, quarterly group meetings	CRC case management program, surgical and clinical oncology departments May 2013–April 2014 (1 year)	EHR review, n = 1613 medical records reviewed, n = 341 assessed for eligibility, n = 229 deemed eligible n = 223 randomized Group A—Diet and PA (n = 55) Group B—Diet only (n = 56) Group C—PA only (n = 56) Group D—control (n = 56)	No, 99.6% of accrual target	Methods: Flexible scheduling, travel allowance 6 months: Total: 95% (n = 212) Group A 96.4% (n = 53) Group B 92.9% (n = 52) Group C 96.4% (n = 54) Group D 94.6% (n = 53) 12 months: Total: 91.9% (n = 205) Group A 94.5% (n = 52) Group B 85.7% (n = 48) Group C 92.9% (n = 52) Group D 94.6% (n = 53) 18 months: Total: 88.8% (n = 198) Group A 89.1% (n = 49) Group B 83.9% (n = 47) Group C 92.9% (n = 52) Group D 89.3% (n = 50) 24 months: Total: 86.1% (n = 192) Group A 85.5% (n = 47) Group B 82.1% (n = 46) Group C 92.9% (n = 52) Group D 85.7% (n = 48)	No, expected 10% total attrition. Met goal at year 1 but not year 2
Optimune trial, Holtdirk, 2020 (Germany) [32]	Two-arm RCT Target accrual: n = 360 (1) BC (2) diagnosed within 5 years (3) completed primary treatment >1 month before enrollment	49.9 years (8.2), range: 30–70 years 100% female Not reported 62.3% (n = 226) ≥ some college	Diet, PA, psychosocial support, sleep 6 months	Internet-based	Online (Google ads, internet forums, Facebook), treatment centers, patient associations, support groups, health insurance companies October 2018–April 2020 (1.5 years)	Potential participants indicated their interest using a survey on the study webpage after hearing about the study from one of the recruitment sources, n = 749 showed interest in the study, n = 609 provided consent and were screened, n = 363 deemed eligible n = 363 randomized (n = 181 intervention, n = 182 control)	Yes, 100.8% of accrual target	Methods: Personalized recommendations, responsive web-design approach 3 months: Total: 84.3% (n = 306) Intervention: 77.9% (n = 141) Control: 90.7% (n = 165) 6 months: Total: 72.4% (n = 263) Intervention: 69.6% (n = 126) Control: 75.3% (n = 137)	No retention goal identified
Kanker Nazorg Wijzer (Cancer Aftercare Guide) Kanera, 2016 (Netherlands) [33]	Two-arm RCT Target accrual: n = 376 (1) cancer diagnosis (2) completed primary treatment between 4 and 56 weeks previous	Intervention: 55.6 (11.5) years; Control: 56.2 (11.3) years 80% female (n = 369), 20% male (n = 93) Not reported 27.6% (n = 143) ≥ some college	Diet, PA, smoking cessation, psychosocial support 6 months	Home-based, online modules	Hospitals, outpatient clinics November 2013 to June 2014 (8 months)	EHR review, n = 1303 assessed for eligibility, n = 492 completed baseline, n = 30 did not complete informed consent, n = 475 deemed eligible, n = 240 eligible in control, n = 252 intervention n = 518 randomized (n = 253 control, n = 265 intervention)	Yes, 137.8% of accrual target	Methods: Email reminders 6 months: Total: 79% (n = 409) Intervention: 70.9% (n = 188) Control: 87.4% (n = 221)	Yes, expected 20–23% attrition
PRO-MAN trial, Mohamad, 2019 (Scotland) [34]	Two-arm RCT A priori target accrual not defined (1) PC (2) diagnosis within the previous 36 months	65.5 (5.6) years 100% male Not reported Not reported	Diet and PA 12 weeks Extended follow-up at 6 and 12 months	Group meeting, pedometer, telephone-based dietary advice, online resources	Urological Cancer Database October 2013–December 2013 (3 months)	EHR review, n = 286 screened, n = 92 assessed for eligibility, n = 63 deemed eligible n = 62 randomized (n = 31 intervention, n = 31 control)	Not reported	Methods: Phone calls 12 weeks: Total: 87.1% (n = 54) Intervention: 83.9% (n = 26) Control: 90.3% (n = 28) 6 months: Total: 82.3% (n = 51) Intervention: 77.4% (n = 24) Control: 87.1% (n = 27) 12 months: Total: 43.5% (n = 27) Intervention: 35.5% (n = 11) Control: 51.6% (n = 16)	No goal identified
Efficacy of a diet and physical activity intervention for patients on ADT, O’Neill, 2015 (North Ireland) [35]	Two-arm RCT Target accrual: n = 94 (1) PC (2) planning to or are already being treated with LHRHa for 6 months	Intervention: 69.7 (6.8) years; Control: 69.9 (7.0) years 100% male Not reported 36.2% (n = 34) ≥ some college	Diet and PA 6 months	Home-based meetings with a nutritionist, guidebook, pedometer, involved partner or caregiver if possible	Cancer center at Belfast City Hospital August 2009–March 2011 (1.5 years)	Clinical recruitment, n = 158 patients deemed eligible, n = 94 enrolled, 59.5% recruitment rate n = 94 randomized (n = 47 intervention, n = 47 control)	Yes, 100% of accrual target	Methods: Provision of intervention materials to control group at the end of the trial, telephone calls to monitor participant progress 3 months: Total: 96.8% (n = 91) Intervention: 95.7% (n = 45) Control: 97.9% (n = 46) 6 months: Total: 95.7% (n = 90) Intervention: 95.7% (n = 45) Control: 95.7% (n = 45)	Yes, expected 30% attrition
Mail-based lifestyle interventions for BC survivors, Park, 2016 (United States) [36]	Three-arm RCT Target accrual: n = 225 (1) BC, stage I–II (2) post-primary treatment in the previous 3 months (3) eligibility expanded to stage 0–II with diagnosis in the previous 1.5 years due to recruitment difficulties	56.4 years (TTMI: 55.7 (10.9) years, SLM: 57.7 (10.7) years, Control: 55.7 (10.9) years) 100% female 94.2% (n = 163) White 87.2% (n = 151) ≥ some college	Diet and PA 4 months Extended follow-up at 7 months	Home-based: mailed materials with 5 or more f/v per day, <30% calories from lipids, increase to moderate PA 150 min/week.	Cancer centers, flyers, hospitals, ClinicalTrials.gov, mailings September 2011–October 2013 (2 years)	85% recruited through cancer center registry, 7.5% through ClinicalTrials.gov and a regional hospital, 6.4% through mailings, and 1.2% through flyers, n = 2279 identified as potentially eligible, n = 177 screened, n = 173 deemed eligible n = 173 randomized TTMI n = 57 SLM n = 58 Control n = 58	No, 76.9% of accrual target Reason: funding period ended	Methods: USD 60 compensation for completion of 4- and 7-month follow-up, provision of intervention materials to control group at the end of the trial 4 months: Total: 76.9% (n = 133) TTMI 70.2% (n = 40) SLM 74.1% (n = 43) Control 86.2% (n = 50) 7 months: Total: 75.7% (n = 131) TTMI 75.4% (n = 43) SLM 70.7% (n = 41) Control 81% (n = 47)	No retention goal identified
Rx for Better Breast Health Trial, Ramirez, 2017 (United States) [37]	Two-arm RCT Target accrual: n = 150 (1) BC (2) ≥2 months post-treatment	56.6 years (intervention: 55.3 (9.8) years, control: 57.9 (8.8) years) 100% female 43.1% (n = 66) White 85.6% (n = 131) ≥ some college	Diet 6 months Extended follow-up at 12 months	In-person nutrition workshops, telephone calls with patient navigators, newsletters, motivational interviewing	Not reported Not reported	No details on recruitment methods, n = 301 screened for eligibility, n = 180 deemed eligible n = 153 randomized (n = 76 intervention, n = 77 control)	Yes, 102% of accrual target	Methods: Phone call to follow-up after a missed session and electronic copies of all workshop materials, tailored newsletter Total: 81.7% (n = 125) Intervention: 78.9% (n = 60) Control: 84.4% (n = 65)	No retention goal identified
Living Well after Breast Cancer, Reeves, 2017 (Australia) [42]	Two-arm pilot feasibility RCT Target accrual: n = 90 (1) BC, stage I-III (2) diagnosed in the previous 9–18 months (3) BMI 25–40 kg/m^2^	55.3 (8.7) years 100% female 96.7% (n = 87) White 67.8% (n = 61) ≥ some college	Diet and PA 6 months	Telephone-based calls with coaches, pedometer, self-monitoring diary, workbook, digital scale, calorie counter book	State-based cancer registries, with oncologist permission to contact October 2010 to February 2012 (2.3 years)	Cancer registry, n = 927 potentially eligible participants identified, n = 743 participants given oncologist consent and sent study letters, n = 248 participants consented to contact, n = 213 assessed for eligibility n = 90 randomized (n = 45 intervention, n = 45 control)	Yes, 100% of accrual target	Methods: Individualized coaching Total: 82.2% (n = 74) Intervention: 88.9% (n = 40) Control: 75.6% (n = 34)	No, expected 10% attrition
Exercise and Nutrition to Enhance Recovery and Good Health for You (ENERGY) Trial, Rock, 2013; and Sedjo, 2016 (United States) [38,39]	Two-arm RCT Target accrual: n = 690 (1) BC, stage I–III (2) diagnosed in the previous 5 years	Intervention: 56.1 (9.4) years, control: 56.5 (9.5) years 100% female 78.3% (n = 546) White 85.1% (n = 593) ≥ some college	Diet and PA 24 months	In-person, group sessions, individualized telephone/email contact to support study goals, print newsletters	Local or regional cancer registries, clinics, television, radio, local print media, community support groups, local events or organizations Fall of 2010–May 2012 (1.5 years)	Potentially eligible participants identified from the cancer registry were mailed study invitations or participants called study staff from distributed flyers, n = 11,311 tumor registry or oncology referral letters sent, n = 2740 flyers distributed, n = 7501 telephone contacts or records reviewed, n = 5027 breast cancer cases screened, n = 714 baseline visits completed n = 697 randomized (n = 348 intervention, n = 349 control)	Yes, 101% of accrual target	Methods: Monthly standardized contacts, invitation to bi-monthly seminars, mailing personalized cards, distribution of donated items (i.e., massage vouchers, coupons, etc.), monthly newsletters Total: 84.2% (n = 587) Intervention: 86.2% (n = 300) Control: 82.2% (n = 287)	No, expected 10% attrition
Living Well After Breast Cancer, Terranova, 2022 (Australia) [10]	Two-arm RCT Target accrual: n = 156 (1) BC, stage II–III (2) diagnosis within 2 years (3) post-primary treatment (4) BMI 25–45 kg/m^2^	55.4 (9.2) years 100% female 98.1% (n = 159) White 59.8% (n = 95) ≥ some college	Diet and PA 18 months	Telephone-based calls with coaches, pedometer, self-monitoring diary, workbook, digital scale, calorie counter book	Hospitals in Brisbane and state cancer registry October 2012–December 2014 (2 years)	Cancer registry, n = 1040 identified, 394 contacted, 170 ineligible and 65 declined, 159 consented and were randomized n = 159 randomized (n = 79 intervention, n = 80 usual care)	Yes, 101.9% of accrual target	Methods: Rapport building, flexible scheduling, travel and parking reimbursement, brief feedback on assessments and newsletter, birthday cards, emergency contact collection 6 months: Total: 89.9% (n = 143) Intervention: 92.4% (n = 73) Control: 87.5% (n = 70) 12 months: Total: 81.8% (n = 130) Intervention: 88.6% (n = 70) Control: 75% (n = 60) 18 months: Total: 80.5% (n = 128) Intervention: 86.1% (n = 68) Control: 75% (n = 60)	Yes, expected 20% attrition at 12 months
Survivor Choices for Eating and Drinking (SUCCEED) Trial, Van Blarigan, 2020 (United States) [40]	Two-arm, wait-list control RCT Target accrual: n = 50 (1) CRC (2) not being treated with chemotherapy (3) considered disease-free or have stable disease	55 years, IQR: 55–62 years 64% female (n = 33), 34% male (n = 17) 70.0% (n = 35) White 96.0% (n = 48) college degree	Diet 12 weeks Extended follow-up at 24 weeks	Web-based intervention, text messaging	University of California San Francisco (UCSF) gastrointestinal oncology clinic, clinics, support groups, national conference April 2017–May 2018 (1 year)	Interested individuals were asked to complete a screening survey online and have a provider complete a form to verify clinical information, n = 94 assessed for eligibility, n = 19 ineligible, n = 11 refused, n = 14 did not complete enrollment procedures n = 50 randomized (n = 25 intervention, n = 25 control)	Yes, 100% of accrual target	Methods: Provision of intervention materials to control group at the end of the trial, text prompts for participation 12 weeks: Total: 90% (n = 45) Intervention: 88% (n = 22) Control: 92% (n = 23) 24-weeks: Total: 84% (n = 42) Intervention: 88% (n = 22) Control: 80% (n = 20)	Yes, expected 20% attrition

RCT—randomized controlled trial; BC—breast cancer; PC—prostate cancer; CRC—colorectal cancer; PA—physical activity; f/v—fruits and vegetables; BMI—body mass index; TTMI—targeting the teachable moment; SLM—standard lifestyle management, EHR—electronic health record.

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
