# Peer review of "Recruitment and Retention Strategies Used in Dietary Randomized Controlled Interventions with Cancer Survivors: A Systematic Review"

_cancers, 2023, doi:10.3390/cancers15174366_

Round 1

Reviewer 1 Report

This is a very well-planned and constructed systematic review. I enjoyed reading this manuscript and believe that the aims are clearly described, the methodology is appropriate, the results are clearly described, and the discussion provides good insights into the results.

I have one very minor correction that needs to be made.

The following sentence is grammatically incorrect “. Compared to studies which did not meet their identified retention goals, studies that more commonly included delayed delivery of intervention materials to the control group, rapport building and personalized messaging, or tailored materials.”

Please revise this sentence. 

Author Response

Response to Reviewer 1 Comments

Point 1: I have one very minor correction that needs to be made. The following sentence is grammatically incorrect “. Compared to studies which did not meet their identified retention goals, studies that more commonly included delayed delivery of intervention materials to the control group, rapport building and personalized messaging, or tailored materials.”

Please revise this sentence.

Response 1: Thank you for your comment. The sentence has now been changed to: “Compared to studies which did not meet their identified retention goals, studies that did meet their retention goals more commonly included delayed delivery of intervention materials to the control group, rapport building and personalized messaging, or tailored materials.”

Reviewer 2 Report

Dear author,

Thank you for sharing your research.

I found that it has been elaborated with rigorous and precise methods.

The concluding admonition should guide future researchers in conducting studies on the same topic, especially if the duration of the observation period is particularly long (greater than 12 months).

No changes are necessary.

Best regards.

Author Response

Response to Reviewer 2 Comments

Point 1: I found that it has been elaborated with rigorous and precise methods.

The concluding admonition should guide future researchers in conducting studies on the same topic, especially if the duration of the observation period is particularly long (greater than 12 months). No changes are necessary.

Response 1: Thank you for your comments. We have added to this sentence in the conclusion to emphasize this point: “Efforts to improve retention reporting should be enhanced, especially for trials of longer duration.”

Reviewer 3 Report

As a patient who has survived kidney cancer resection surgery for two years, I am very pleased to see an article published that suggests patient planning from a dietary perspective. Although there are many factors that affect the dietary regulation of cancer patients as mentioned by the authors, and it is difficult to reach a consensus so far, it is undeniable that dietary planning is very important for cancer survivors at all stages. Therefore, I am very willing to recommend this article for publication. There are some small questions for discussion and reference only:

1. Can we not only consider the direct stimulation of food and beverage on cancer recurrence, but also consider whether nutrition can promote postoperative recovery and the sequelae of chemotherapy and radiation therapy during the discussion?

2. Whether it is necessary to consider the impact of reasonable food on the patient's mood, which is an important factor affecting rehabilitation.

3. Patients of different age groups have different feedback on the same meal. Do you need to discuss this.

Author Response

Response to Reviewer 3 Comments

Point 1: Can we not only consider the direct stimulation of food and beverage on cancer recurrence, but also consider whether nutrition can promote postoperative recovery and the sequelae of chemotherapy and radiation therapy during the discussion?

Response 1: This is a great question but is outside the scope of this review. We felt that studies which enrolled cancer patients currently undergoing treatment would utilize very different methods for recruitment and retention than for cancer survivors who are post-treatment and may not be attending clinic visits regularly. Patients currently undergoing treatment also have different symptom burden that may impact their participation in a trial and this would vary depending on their treatment modality. This review focuses on cancer survivors post-treatment to capture the methods used to engage a population that we know can benefit from these behavior modifications but which may not be engaging in clinical care regularly anymore and may be more challenging to engage in an intervention or program.

Point 2: Whether it is necessary to consider the impact of reasonable food on the patient's mood, which is an important factor affecting rehabilitation.

Response 2: This is a valid point, but not something that was explored in this review. We have added it as a point for future research to consider in the discussion:

“Future research should also explore other factors that may impact survivor engagement and adherence to an intervention such as affect, dietary preference, and age. Clear reporting of patient-reported outcomes is necessary to understand if attrition and resulting missing data are related to poorer outcomes, methodological issues, or lack of interest in the intervention.”

Point 3: Patients of different age groups have different feedback on the same meal. Do you need to discuss this.

Response 3: We did not look at acceptability of the intervention based on age, but this would be interesting to consider. We have included this as a point for future research in the discussion:

“Future research should also explore other factors that may impact survivor engagement and adherence to an intervention such as affect, dietary preference, and age. Clear reporting of patient-reported outcomes is necessary to understand if attrition and resulting missing data are related to poorer outcomes, methodological issues, or lack of interest in the intervention.”

Reviewer 4 Report

This is an interesting paper about cancer and diet.

Overall, the article is well done, some problems should be fixed. The Discussion section should be redone, actually sound like a Discussion, so the authors should make the changes in this section. Finally, the conclussion is confuse, please answer to the aim correctly.

Author Response

Response to Reviewer 4 Comments

Point 1: Overall, the article is well done, some problems should be fixed. The Discussion section should be redone, actually sound like a Discussion, so the authors should make the changes in this section.

Response 1: Thank you for your comment. We have modified the discussion so that it is more clear. In short, we added more clear references to previous work and how it compares to our findings. We also added a more clear statement of guidance for future research. The current outline of the discussion as written is:

  1. Brief summary of the paper’s main conclusions
  2. Comparison of our recruitment related results with previous work
  3. Comparison of our retention related results with previous work
  4. Implications of this work and suggested resources to improve study methodology
  5. Strengths
  6. Limitations
  7. Future directions

Point 2: Finally, the conclussion is confuse, please answer to the aim correctly.

Response 2: We have modified that conclusion to more directly reflect the identified aims. We have added brief summaries of the recruitment and retention methods that were found to be effective. It now reads:

“Achieving and retaining study participants in dietary intervention trials among cancer survivors are important to the interpretation of study findings. The most common recruitment methods utilized for studies which achieved their a priori accrual goals included clinical recruitment and provider referral or recruitment from a cancer registry. Studies with a duration of six months or less and studies with remote or hybrid delivery models were also most likely to have met their accrual goals. Our findings suggest investigators are meeting reporting guidance for transparency in study recruitment; however, reporting of retention methods and rates is less consistent, raising concern as to the interpretation of study findings. Studies which met their identified retention goal most frequently reported regular contact and interaction between study staff and participants. This indicates that rapport building is an important factor for survivor engagement and continuation with a trial. Efforts to improve retention reporting should be enhanced, especially for trials of longer duration.”

Round 2

Reviewer 1 Report

Well done on producing a good-quality paper.

Reviewer 4 Report

Outsanding